# Exploring the mediating effect of personality traits in the relationship between entrepreneurial intentions and academic performance among students

**Smita Panda**[ID], **Vasumathi Arumugam**[ID]*

VIT Business School, Vellore Institute of Technology, Vellore, Tamil Nadu, India

* avasumathi@vit.ac.in

**Data Availability Statement:** All relevant data are within the paper and its Supporting Information files.

## Abstract

This study explores the mediating effect of personality traits in the relationship between entrepreneurial intentions and student academic performance. The sample comprised 175 students from a top-ranked Tamil Nadu, India university. Data was collected using a survey questionnaire as the research instrument. A descriptive research design was employed to understand the variables under investigation comprehensively. Partial Least Squares Structural Equation Modeling (PLS-SEM) and SPSS v25 was utilized as the statistical analysis tool. This study used the Theory of Planned Behaviour as a theoretical framework to explore the mediating effect of personality traits in the relationship between entrepreneurial intentions and academic performance among university students. The study's findings revealed essential insights into the relationship between entrepreneurial intentions, personality traits, and academic performance. The results showed that personality traits significantly mediate the relationship between entrepreneurial intentions and academic performance. This finding suggests that a student's personality traits influence the impact of entrepreneurial intentions on academic performance. Furthermore, the study found that while entrepreneurial intentions did not have a significant direct effect on academic performance, they did have a substantial indirect effect through personality traits. This indicates that personality traits act as a crucial mechanism through which entrepreneurial intentions can influence academic performance among students.

## 1 Introduction

Entrepreneurship has emerged as a crucial catalyst of economic growth, innovation, and job creation. It has been recognized as an essential area of research in business and management [1–3]. Entrepreneurial intentions refer to individuals' actions, attitudes, and activities in pursuing entrepreneurial goals and opportunities [4–6]. It encompasses the behavioral aspects of entrepreneurship, including decision-making processes, risk-taking propensity, proactiveness, innovation, and opportunity recognition exhibited by entrepreneurs and individuals engaged

**Funding:** The authors received no specific funding for this work. The institute will pay the article processing charge.

**Competing interests:** The authors have declared that no competing interests exist.

in entrepreneurial activities [7]. Personality traits are significantly related to entrepreneurial intentions, predicting behaviors of entrepreneurship [8]. Personality traits are enduring patterns of thoughts, feelings, and behaviors that shape individuals' responses to different situations [9]. In the context of entrepreneurial intention, several studies have examined how specific traits relate to the likelihood of engaging in entrepreneurial activities [10, 11].

On the other hand, academic performance is a significant concern for students, educational institutions, parents, and society [12]. Academic performance predicts future success and contributes to developing a strong and competitive workforce [13]. Therefore, understanding the association between personality traits, entrepreneurial intentions, and academic performance can provide valuable insights into how to better prepare students for entrepreneurship and academic success.

Many researchers identified a positive connection between personality traits and entrepreneurial intentions [14]. However, it is still being determined whether the extent to which personality traits mediate the association between entrepreneurial intentions and academic performance is less studied. Very scant studies suggested that personality traits may influence the relationship between entrepreneurial intentions and academic performance [15]. Still, more empirical evidence is needed to support this claim. The mediating role of personality traits between entrepreneurial intention and student academic performance is an emerging area of research. While these two domains might seem distinct, personality traits provide a potential link. Entrepreneurial intention, driven by traits like self-confidence and risk-taking, can influence academic performance through motivational factors, learning strategies, and time management [16]. The present study aims to fill this research gap by exploring the mediating effect of personality traits in the relationship between entrepreneurial intentions and academic performance among university students. This study is significant as it will provide a more comprehensive understanding of the factors influencing university students' academic performance and entrepreneurial intentions.

Understanding the interplay among personality traits, entrepreneurial intentions, and academic performance is important in entrepreneurship and education [17, 18]. To address the existing research gap, this study aims to investigate the mediating role of personality traits in the relationship between entrepreneurial intentions and academic performance among university students. Building upon the Theory of Planned Behavior foundation, this research explores how personality traits may shape the association between entrepreneurial intentions and academic performance. The findings of this study will not only contribute to the existing body of knowledge but also offer practical implications for educational institutions and policymakers, guiding them in designing effective strategies to enhance students' entrepreneurial intentions and academic achievements.

### 1.1 Research questions

RQ1. Do entrepreneurial intentions impact the student's academic performance?

RQ2. Do personality traits mediate the relationship between entrepreneurial intentions and students' academic performance?

RQ3. Does the financial background of the students have an impact on their entrepreneurial intentions?

### 1.2 Research objective

1. To investigate how students' entrepreneurial intentions affect their academic performance.

2. To analyze whether personality traits mediate the relationship between entrepreneurial intentions and students' academic performance.

3. To ascertain the potential influence of students' pocket money on shaping their entrepreneurial intentions.

## 2 Literature review

### 2.1 Entrepreneurial intention

Entrepreneurial intention received considerable attention in the literature as a predictor of behaviors related to entrepreneurship [19, 20]. University students represent an essential group for studying entrepreneurial intention due to their potential to become entrepreneurs in the future [21, 22]. Several studies have identified various factors that influence entrepreneurial intention among university students. These factors include personal characteristics such as gender, age, prior experience with entrepreneurship, and environmental factors such as social norms, cultural values, and economic conditions [23].

Personal characteristics such as self-efficacy, risk-taking propensity, and innovativeness positively related to entrepreneurial intention among university students [24]. Similarly, exposure to entrepreneurship education and prior entrepreneurial experience positively related to entrepreneurial intention among university students [25].

While entrepreneurial intention is a necessary precursor to behaviors related to entrepreneurship, not all individuals with entrepreneurial intentions engage in entrepreneurial activities. The relationship between entrepreneurial intention and behavior is the subject of considerable debate in the literature. A model of entrepreneurial intentions was proposed by eminent researchers that includes two distinct pathways: the attitude-behavioral intention pathway and the subjective norm-behavioral intention pathway [26]. The attitude-behavioral intention pathway posits that entrepreneurial intention leads directly to behaviors related to entrepreneurship. In contrast, the subjective norm-behavioral intention pathway suggests that social norms and external pressures influence behaviors related to entrepreneurship. Individual factors such as self-regulation and self-control may moderate the relationship between entrepreneurial intention and behavior [27]. Moreover, the relationship may vary across different contexts, such as the type of entrepreneurial activity and the level of support from the external environment [28].

Policymakers and educators have recognized the importance of promoting entrepreneurship among university students to foster economic growth and innovation [29, 30]. Accordingly, many universities implemented entrepreneurship education programs to foster entrepreneurial intention and behavior among students. Research has shown that entrepreneurship education promotes entrepreneurial intention and behavior among university students [31]. However, the effectiveness of such programs may be moderated by individual and contextual factors such as prior experience, personality traits, and cultural values [32, 33].

Entrepreneurial intention among university students is a complex phenomenon influenced by various individual and contextual factors [34–36]. While entrepreneurial intention is a necessary precursor to behaviors related to entrepreneurship, the relationship between the two could be more complex and influenced by individual and contextual factors [37]. Promoting entrepreneurship among university students has important implications for education and policy, and entrepreneurship education programs may be effective in fostering entrepreneurial intention and behavior among students.

## 2.2 Student academic performance

Academic performance is an essential outcome for university students [38, 39] and is the subject of extensive research in the literature. Several factors include personal characteristics such as intelligence, motivation, and self-regulation and environmental factors such as social support, teaching quality, and academic culture [40, 41]. Richardson et al. (2012) found that motivation and self-regulation positively related to academic performance among university students. However, Kuh et al. (2006) found that social support and academic culture were significant predictors of academic performance.

Academic performance is related to various other outcomes, such as career success, health, and well-being [42]. Research has shown that academic performance is positively associated with career success, earnings [43], and physical and mental health [44]. Moreover, academic performance may be influenced by factors outside the academic domain, such as extracurricular activities and part-time work [45]. They have also found that part-time work negatively affects academic performance among university students.

Policymakers and educators have recognized the importance of promoting academic performance among university students to foster human capital and economic growth. Accordingly, many universities have implemented various policies and programs to improve academic performance among students [46, 47]. Research has shown multiple interventions, such as mentoring, academic support, and students. However, the effectiveness of such interventions may be moderated by individual and contextual factors such as prior academic achievement, socioeconomic status, and institutional culture [48, 49].

Various individual and contextual factors influence academic performance among university students and are related to other outcomes, such as career success and health [50]. Promoting academic performance has important implications for education and policy, and various interventions may effectively improve academic performance among university students [51].

## 2.3 Personality traits

The concept of personality traits has been extensively studied by psychologists, with different models proposed over the years [52–54]. Research has shown that personality traits can influence numerous facets of an individual's life, including academic and work performance, relationships, and health [55]. Understanding the relationship between personality traits and different outcomes can help individuals and organizations make better decisions [56]. Personality traits are an enduring pattern of thoughts, feelings, and behaviors that make an individual unique [57, 58].

The concept of personality traits has been studied extensively by psychologists, with different theories and models proposed over the years [59]. Some of the most influential models include the Myers-Briggs Type Indicator (MBTI) [60], the Five-Factor Model (FFM) [61, 62], and the HEXACO model [63]. Personality traits are an essential factor that can influence various aspects of a university student's life, including risk tolerance, employment preference, need for achievement, locus of control, and entrepreneurial alertness [64].

Personality traits can influence an individual's employment preferences. Students with high openness to experience and conscientiousness were likelier to self-employment than traditional employment [65]. Similarly, other studies have shown that students with high levels of extraversion and risk-taking tend to choose jobs that provide opportunities for creativity and innovation [66]. A person's locus of control is a psychological feature related to their belief in their power to influence their life occurrences [67]. According to research, students with a greater internal locus of control do better academically and are more willing to seek out complex challenges [68].

Similarly, individuals with a high external locus of control may be less likely to take risks and pursue entrepreneurial opportunities. The need for achievement is a personality characteristic that describes a person's ambition to achieve success and mastery in their endeavors [69, 70]. Individuals with a high need for achievement tend to be more persistent and focused [71]. These individuals are also more likely to seek challenging opportunities and have higher entrepreneurial alertness [64, 72, 73]. Risk tolerance is a personality trait that refers to an individual's willingness to take risks [74–76]. According to research, individuals with a high-risk tolerance are more likely to participate in entrepreneurial activities [77, 78].

Similarly, individuals with low-risk tolerance may be more likely to avoid entrepreneurial activities and seek out traditional employment opportunities [79]. Entrepreneurial alertness refers to a person's ability to recognize and identify entrepreneurial opportunities [80, 81]. Research has shown that individuals with higher entrepreneurial alertness are most likely to be engrossed in entrepreneurial activities and pursue entrepreneurial opportunities [82, 83]. Similarly, individuals with low entrepreneurial alertness may be less likely to recognize entrepreneurial opportunities and pursue entrepreneurial activities [84].

Personality traits are essential in influencing various aspects of a university student's life, including their employment preference, locus of control, need for achievement, risk tolerance, and entrepreneurial alertness. Understanding the relationship between these personality traits and different outcomes can help individuals and institutions make better decisions.

## 2.4 Relationship between entrepreneurial intention and personality traits

Several studies have suggested that personality traits are critical in shaping an individual's entrepreneurial intentions [85, 86]. Moreover, research has also indicated that personality traits significantly predict entrepreneurial intention among university students [87]. The locus of control, need for achievement, and risk-taking propensity were significant predictors of entrepreneurial intention among university students in China [88].

Furthermore, several studies have highlighted the significance of entrepreneurial alertness as a personality trait influencing entrepreneurial intention among university students [72]. Entrepreneurial alertness refers to an individual's ability to identify and exploit entrepreneurial opportunities [89]. Entrepreneurial alertness was positively related to entrepreneurial intention among university students in China [90].

In addition to the above, research has shown gender variations in personality characteristics and entrepreneurial intention among university students. Women had lower levels of entrepreneurial intention than men, and the relationship between personality traits and entrepreneurial intention differed between men and women [91].

H1: Entrepreneurial intention and personality traits are significantly related.

## 2.5 Relationship between personality traits and student academic performance

Personality traits have long been studied to predict individual outcomes, including academic performance [92, 93]. University students, in particular, are a unique population with diverse personality traits that can impact their academic performance [94]. This literature review explores the relationship between personality traits and student academic performance among university students. Personality traits have been found to impact employment preference among university students [95]. Locus of control is another personality trait that refers to an individual's belief about how much they can control events. Students with an internal locus of control had higher GPAs than those with an external locus of control [96].

Moreover, the need for achievement is a trait that refers to an individual's desire to excel and achieve goals [97]. Students who scored high on the need for achievement were found to have higher GPAs than those who scored low on this trait. Finally, risk tolerance refers to an individual's willingness to take risks [98]. Students who scored high on risk tolerance were found to have lower GPAs than those who scored low on this trait [99].

H2: There is a relationship between personality traits and student academic performance.

## 2.6 Relationship between entrepreneurial intention and student academic performance

Entrepreneurship has significantly contributed to economic growth, innovation, and job creation in many countries worldwide [100]. Several studies have explored the relationship between entrepreneurial intention and student academic performance. Entrepreneurial intention positively impacted academic performance and found that students with higher entrepreneurial intentions performed better academically than those with lower entrepreneurial intentions [101].

Furthermore, researchers have indicated that students with entrepreneurial intentions had higher GPAs and academic achievements than those without entrepreneurial intentions [102, 103]. However, other studies have reported mixed findings regarding the relationship between entrepreneurial intention and academic performance. There is no significant association between entrepreneurial intention and student's academic performance [91]. Additionally, researchers revealed that although entrepreneurial intention positively affected academic performance, it was not statistically significant [104]. The mixed findings in the literature may be due to several factors, including the sample size, measurement of variables, and the study context. It is also evident that the association between entrepreneurial intention and academic performance was stronger among female students than male students [105]. Moreover, conscientiousness and emotional stability were positively related to academic performance [106]. A study revealed that openness, conscientiousness, and emotional stability were positively associated with academic performance [107].

Furthermore, students' academic performance was positively related to their need for achievement, locus of control, and self-efficacy [108]. A positive relationship between academic performance and risk-taking propensity [109].

H3: There is a significant relationship between entrepreneurial intention and student academic performance.

## 2.7 The mediation role of personality traits

Personality traits have been extensively studied due to their significant impact on various outcomes, including academic performance, job performance, mental and physical health, and social relationships [110]. In recent years, researchers have shown increasing interest in understanding the mediating effect of personality traits, which refers to the idea that personality traits can influence other variables through an intervening variable [111]. Extraversion may lead to increased job performance, but conscientiousness may mediate this relationship [112]. Several well-studied personality traits have been found to have mediating effects. Employment preference mediated the relationship between personality traits and entrepreneurial intentions [113]. This suggests that personality traits such as extraversion and openness to experience lead to a choice for entrepreneurship, influencing entrepreneurial intentions.

Similarly, research has studied that employment preference mediated the relationship between personality traits and entrepreneurial self-efficacy [114]. Conscientiousness and agreeableness led to a preference for traditional employment, resulting in lower levels of entrepreneurial self-efficacy. Locus of control has also been identified as a mediating variable between personality traits and entrepreneurial intentions. Studies have found that personality traits such as extraversion and openness to experience lead to an internal locus of control, influencing entrepreneurial intentions [115]. An extended study of this finding demonstrated that locus of control mediated the relationship between personality traits and entrepreneurial self-efficacy [116]. Specifically, conscientiousness and agreeableness were found to influence an internal locus of control, leading to higher levels of entrepreneurial self-efficacy.

Additionally, the need for achievement is another mediating variable between personality traits and entrepreneurial intentions. Additionally, the need for achievement is another mediating variable between personality traits and entrepreneurial intentions. Personality traits such as extraversion and openness to experience lead to a high need for achievement, subsequently influencing entrepreneurial intentions [117]. Some studies further expanded this finding, revealing that the need for achievement mediates the relationship between personality traits and entrepreneurial self-efficacy [118]. Conscientiousness and agreeableness were associated with a high need for achievement, resulting in higher levels of entrepreneurial self-efficacy. Furthermore, risk tolerance has been identified as a mediating variable between personality traits and entrepreneurial intentions [119]. The study showed that personality traits influence risk tolerance, influencing entrepreneurial intentions.

Numerous studies have demonstrated the mediating effects of various personality traits on different variables related to entrepreneurship. These findings highlight the complex relationships and underlying mechanisms involved in the relationship between personality traits and entrepreneurial outcomes. Understanding these mediating effects can provide valuable insights for individuals, educators, and policymakers interested in promoting behaviors related to entrepreneurship and success.

H4: Personality traits mediate the relationship between entrepreneurial intention and student academic performance.

## 3 Theoretical backgrounds

The theory of Planned Behavior offers a suitable theoretical foundation for exploring the interconnection among entrepreneurial intentions, personality traits, and academic competence [120]. According to this theory, individuals' intentions to engage in a particular behavior, such as entrepreneurship, are influenced by three key factors: attitudes, subjective norms, and perceived behavioral control. Attitudes refer to individuals' thoughts and evaluations of the activity, while subjective norms consider the influence of social factors and the expectations of others. Perceived behavioral control pertains to individuals' beliefs about their ability to carry out the behavior. This study introduces personality traits as potential mediators in the relationship between entrepreneurial intentions and academic success. They are seen as factors that can potentially influence the strength or direction of the relationship between entrepreneurial intentions and academic performance.

By employing the Theory of Planned Behavior as the theoretical framework, this research aims to understand how personality traits may mediate the relationship between entrepreneurial intentions and academic success. The theory systematically examines the factors that shape individuals' intentions and behavior, considering their attitudes, social influences, and self-perceived control. Personality traits that can impact one's attitudes, social norms, and

perceived control are hypothesized to mediate this relationship. By incorporating personality traits as mediators, the study seeks to uncover the underlying mechanisms through which entrepreneurial intentions may influence academic achievement, shedding light on the complex dynamics at play.

## 4 Model framework

Fig 1 explains the conceptual model pertains to the impact of entrepreneurial intentions on the student academic performance with the mediating role of personality traits. Here, attitude towards entrepreneurship, perceived behavioral control, subjective norms, perceived educational support, propensity to act are the most cited dimensions used to describe entrepreneurial intentions [26]. Student's self-efficacy, self-set goals, academic competence, ability and time management explains the student academic performance. Employee preference, locus of control, need for achievement, risk tolerance and entrepreneurial alertness are the most cited dimensions for personality traits [64].

## 5 Methodology

The foundation of this study rests on adopting an explanatory design that draws on quantitative methods and hypotheses. Primary data was gathered through a well-structured survey questionnaire, and subsequent statistical tests were performed meticulously to analyze [121]. The survey questionnaire was distributed amongst students in esteemed universities renowned for their academic excellence.

### 5.1 Type of research

This study aims to investigate a specific phenomenon within a particular time and location, exploring the mediating effect of personality traits in the relationship between entrepreneurial intentions and academic performance among university students. Given the nature of the

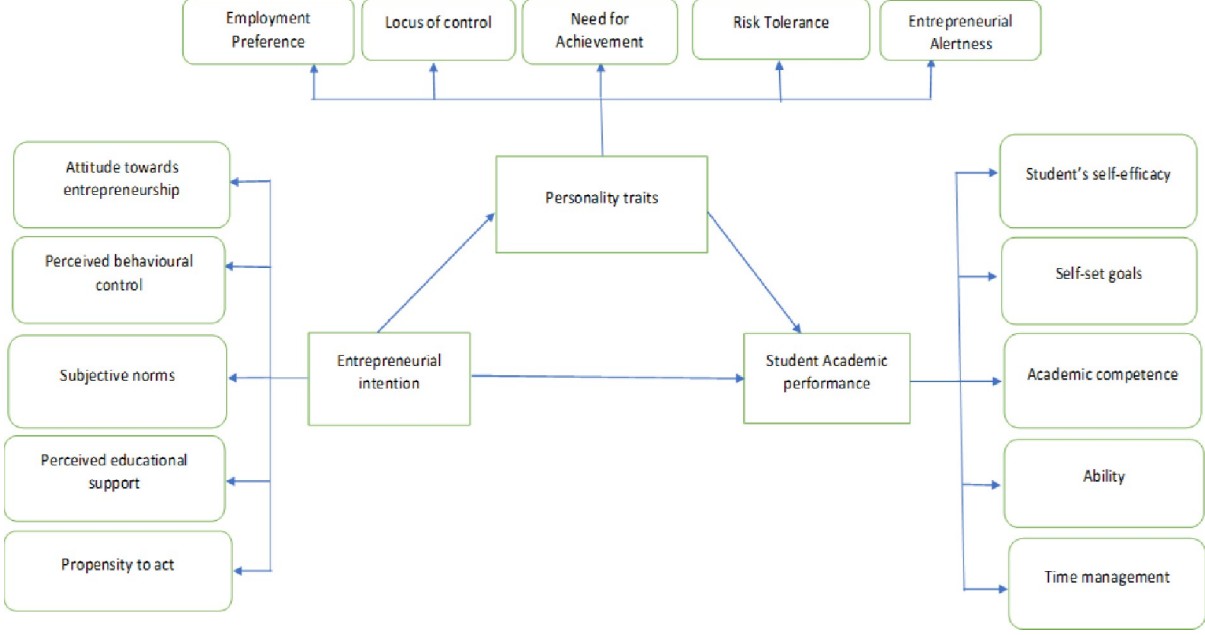

**Fig 1. Conceptual framework of the study.**

research objective and the need to gather comprehensive and detailed information about the variables of interest, the descriptive research design was deemed appropriate for the study.

Descriptive research involves the systematic collection, analysis, and interpretation of data to describe the characteristics of a particular population or phenomenon [122]. This study examines the relationship between entrepreneurial intentions, personality traits, and academic performance among university students. By employing a descriptive research design, the study provides a detailed and accurate portrayal of these variables within the specific time and location under investigation.

One of the primary advantages of using a descriptive research design is its ability to provide a comprehensive overview of the studied variables. This design allows researchers to collect data from various sources, including surveys, interviews, and existing records, enabling a more thorough exploration of the phenomenon [123]. In the case of this study, data was collected through self-report measures of entrepreneurial intentions, personality traits, and academic performance to provide a comprehensive understanding of the relationship between these variables.

Moreover, the descriptive research design enables the researcher to examine the phenomenon naturally without imposing manipulations or interventions [123]. By adopting this approach, the study can capture university students' real-world experiences and behaviors regarding their entrepreneurial intentions, personality traits, and academic performance. This allows for a more authentic representation of the variables under investigation, enhancing the validity and ecological validity of the study findings.

## 5.2 Sample size

Given the challenges of determining the standard deviation and population mean, the researchers implemented a snowball sampling technique to overcome these obstacles. The survey was conducted among students enrolled in a prestigious university in Tamil Nadu, India, known for its high academic standing and student population nationwide. The size of the sample is 175 students in the top-ranked university.

## 5.3 Research instrument

This study is entirely based on empirical evidence, so the researchers exclusively utilized a questionnaire for data collection. The reason for using a questionnaire was its cost-effectiveness and convenience. The questionnaire was divided into two sections: Part I focused on the participant's demographic information like age, gender, educational qualification, pocket money, total number of members in the family, occupation of parents, no. of siblings, and the number of dependents on parents, while Part II comprised questions devised by the researcher to measure entrepreneurial intentions [124], personality traits [64], and academic performance [125].

## 5.4 Data analysis procedures

The data collected through questionnaires were entered and analyzed using Partial Least Squares Structural Equation Modeling (PLS-SEM) and SPSS v25 for statistical analysis. While all scales were conceptually validated separately, the researchers also aimed to ensure their statistical distinctiveness for this study.

This empirical study aims to predict and elaborate on latent variables based on contemporary theory. In recent times, one of the most persuasive techniques that have revolutionized the field is PLS-SEM, widely adopted to test the effectiveness of structural modeling in explaining and evaluating constructs [126]. Furthermore, it is recognized as an adaptive model

assessment technique [127]. PLS-SEM was chosen for this study due to its lower sample size requirement than Amos and data normality requirements [128], enabling the researchers to overcome normality and sample size constraints. The researchers utilized the PLS algorithm and bootstrapping technique to ascertain factor loadings, evaluate construct validity, and assess internal consistency reliability [129]. This technique tests hypotheses by analyzing path coefficients and significance levels. The measurement model is initially computed, followed by an evaluation of the structural model in subsequent assessments [130].

The rationale for adopting the variance-based Partial Least Squares Structural Equation Modeling (PLS-SEM) approach is expounded in this study. The fundamental motivation behind choosing PLS-SEM lies in its capability to simultaneously assess the relationships between latent constructs through the structural model and the associations between indicators and their corresponding latent constructs through the measurement model. This approach comprehensively examines the relationships between variables [131, 132]. Additionally, the variance-based PLS-SEM method was considered suitable for this study as it provides statistically consistent estimates of indirect effects in simple mediation models, employing bootstrapping techniques that utilize standard errors for path coefficients. This ensures robustness in estimating the indirect effects [111, 133–135].

To ensure the reliability and accuracy of the measurement instruments used in this study, the researchers employed various criteria, including Cronbach's alpha, composite reliability (CR), and average variance extracted (AVE). The findings reported in Table 1 demonstrate that the alpha values range from 0.880 to 0.930, while the CR values range from 0.507 to 0.619. These results establish the reliability of the measures, surpassing the threshold of 0.50, thereby confirming their convergent validity [136]. By employing these rigorous criteria, the study ensures the trustworthiness and robustness of the measurement instruments.

Moreover, it is generally recommended that the AVE for each underlying construct exceeds 0.50 [137, 138]. As illustrated in Table 1, the AVE values for each latent construct surpass the 0.50 threshold, providing further evidence of satisfactory convergent validity [139].

**5.4.1 Measurement model.** The standardized factor loadings indicate the strength of the relationship between each observed variable and its corresponding latent construct. In Table 1, the observed variable "SAPTM7" has a factor loading of 0.879, strongly related to the latent construct "student academic performance."

*5.4.1.1 Construct reliability and validity.* Cronbach's alpha is a measure of internal consistency, which indicates how well the items in a scale measure the same construct [140]. Composite reliability is a measure of overall reliability that considers the intercorrelations between all items on a scale [141]. The average variance extracted (AVE) is a measure of convergent validity, which indicates how much variance in the construct is captured by the items in the scale.

As indicated in Table 2, three variables have Cronbach's alpha and composite reliability scores above the recommended threshold of 0.7, indicating good internal consistency. The AVE scores for all three variables are also above the recommended threshold of 0.5, indicating good convergent validity. The results of this analysis suggest that the three variables, EI, PT, and SAP, are all reliable and valid measures of their respective constructs.

*5.4.1.2 Discriminant validity.* The measures' discriminant validity was evaluated using the Fornell-Larcker criterion, as presented in Table 3. Discriminant validity is established when each latent construct's average variance explained (AVE) exceeds the squared correlation with any other construct [136]. Table 3 compares the square root of the AVEs and the squared correlations between the latent variables with the values in boldface. As indicated in the table, the AVE for each latent construct exceeded the squared correlation with any other construct, confirming the presence of satisfactory discriminant validity [136].

**Table 1. Results of the measurement model.**

|  | EI | PT | SAP |
|---|---|---|---|
| PTEA3 |  | 0.799 |  |
| EIATE2 | 0.629 |  |  |
| EIATE3 | 0.741 |  |  |
| EIPBC4 | 0.696 |  |  |
| EIPBC5 | 0.654 |  |  |
| EIPBC6 | 0.746 |  |  |
| EIPES1 | 0.756 |  |  |
| EIPES2 | 0.729 |  |  |
| EISN1 | 0.699 |  |  |
| EISN4 | 0.624 |  |  |
| EISN5 | 0.649 |  |  |
| PTNFA2 |  | 0.649 |  |
| PTEA6 |  | 0.689 |  |
| PTEP1 |  | 0.710 |  |
| PTEP6 |  | 0.745 |  |
| PTLOC2 |  | 0.800 |  |
| PTLOC5 |  | 0.621 |  |
| PTNFA3 |  | 0.674 |  |
| PTRT2 |  | 0.681 |  |
| PTRT4 |  | 0.727 |  |
| SAPAB2 |  |  | 0.814 |
| SAPAB4 |  |  | 0.858 |
| SAPAC1 |  |  | 0.685 |
| SAPAC2 |  |  | 0.608 |
| SAPSSE2 |  |  | 0.808 |
| SAPSSE3 |  |  | 0.810 |
| SAPSSG1 |  |  | 0.791 |
| SAPSSG3 |  |  | 0.779 |
| SAPTM1 |  |  | 0.802 |
| SAPTM7 |  |  | 0.879 |

Table 3 reveals that the AVE for each variable was greater than all squared correlation values with other constructs, based on the recommendation given by the study [136]. This finding indicates that the measurement instruments used in the study are distinct and appropriate for assessing the intended model [139].

Fig 2 illustrates the initial model that served as a guide to demonstrate the relationship between impact of the entrepreneurial intentions on the student academic performance. The model is composed of three latent variables and 30 indicators. The observed values shown for each indicator were measured using the items included in the questionnaire.

**Table 2. Construct reliability and validity.**

|  | Cronbach's Alpha | Composite Reliability | Average Variance Extracted (AVE) |
|---|---|---|---|
| EI | 0.880 | 0.902 | 0.581 |
| PT | 0.891 | 0.911 | 0.507 |
| SAP | 0.930 | 0.942 | 0.619 |

**Table 3. Discriminant validity.**

| Fornell-Larcker Criterion | | | |
| --- | --- | --- | --- |
| | EI | PT | SAP |
| EI | **0.694** | | |
| PT | 0.585 | **0.712** | |
| SAP | 0.622 | 0.608 | **0.787** |

**5.4.2 Structural model.** The structural model is described in the Fig 3 and the following analysis has been done based on the model.

*5.4.2.1 Hypotheses testing.* To assess the hypotheses, the present study used the bootstrapping technique, a resampling method involving iteratively drawing samples from the original dataset 5,000 times. The acceptance or rejection of the hypotheses was determined based on various statistical measures, including the t-value, p-value, and bias-corrected confidence interval. The bootstrap resampling technique was employed to evaluate the hypotheses' significance further.

Upon examining the results presented in Table 4, it is evident that the hypotheses of the relationship between entrepreneurial intention and personality traits, as well as personality traits and student academic performance, were supported. This finding suggests a statistically significant mediating effect of personality traits in the relationship between entrepreneurial intention and student academic performance. The coefficient (β) value of 0.738, the t-value of 5.231, and the p-value of 0.000 provide empirical evidence to support this conclusion.

Utilizing the bootstrapping technique in this study allowed for a rigorous examination of the hypotheses. The results indicated a significant mediating role of personality traits in the relationship between entrepreneurial intention and student academic performance. These findings contribute to understanding the complex interplay between entrepreneurial intention, personality traits, and student academic performance.

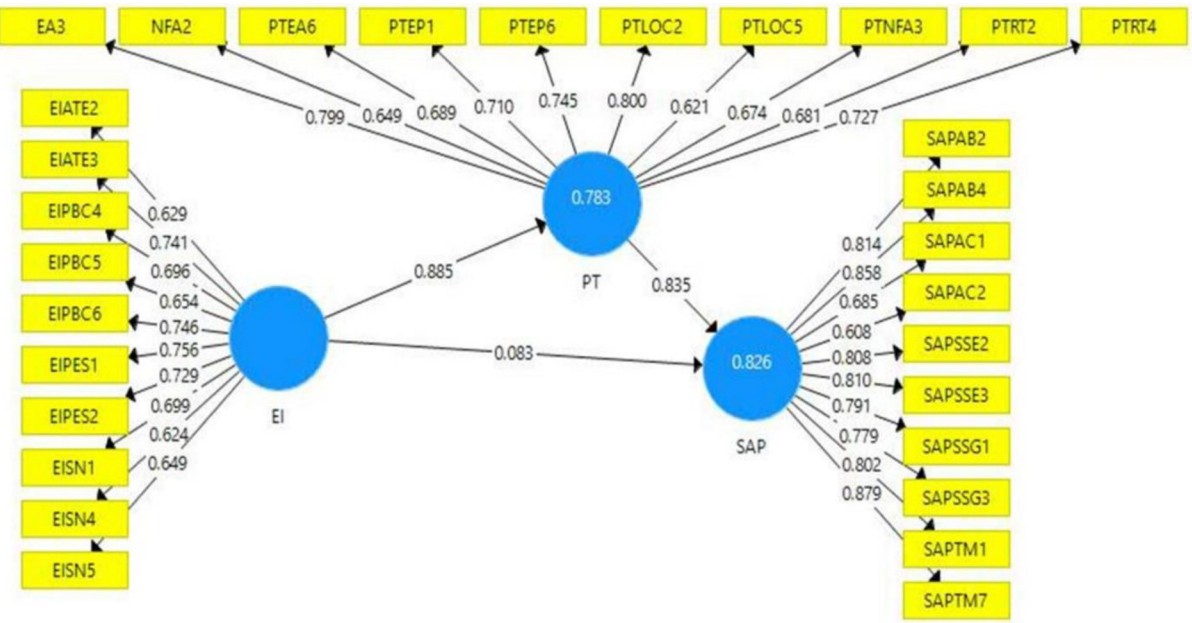

**Fig 2. Measurement model.**

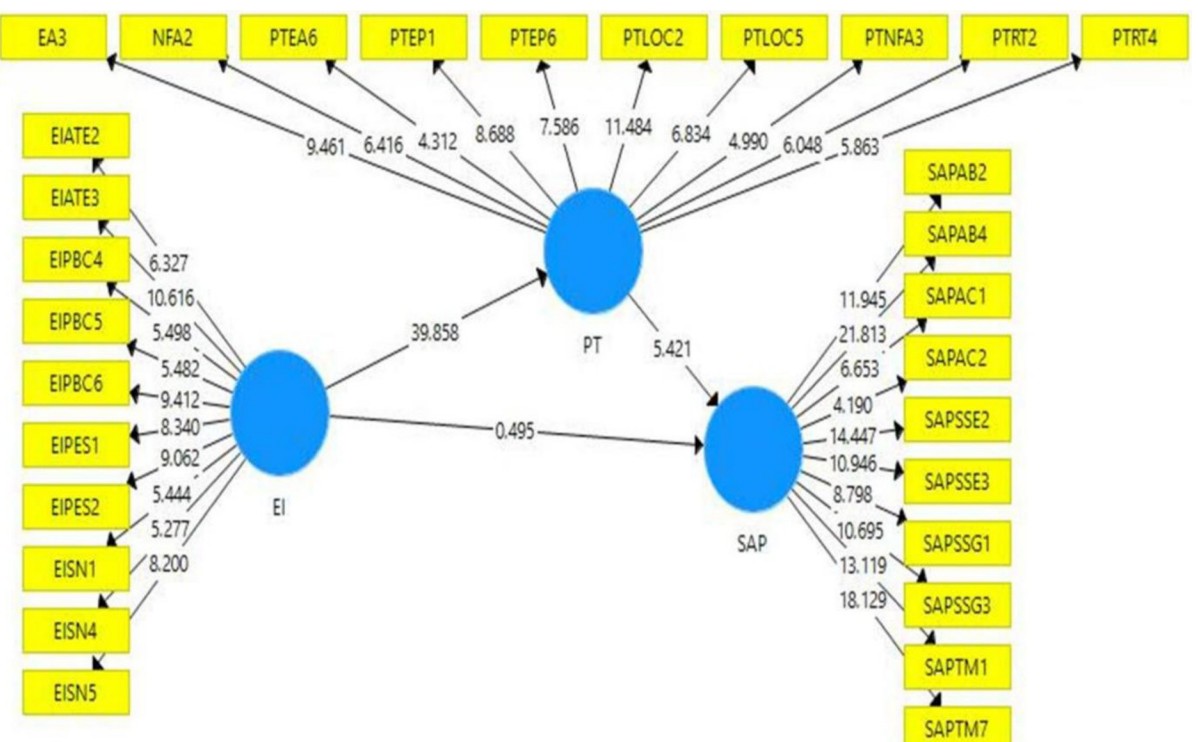

**Fig 3. Structural model.**

*5.4.2.2 $Q^2$ predictive relevance.* The $Q^2$ analysis provides insights into the predictive relevance of a model. A $Q^2$ value greater than 0 indicates the model's good predictive relevance. In the given analysis, we can refer to Table 5 to examine the $Q^2$ values associated with personality traits and student academic performance. The table shows that the Q square value for EI is 0.000, meaning that the variable has no predictive power; the $Q^2$ value for personality traits is calculated to be 0.366, while the $Q^2$ value for student academic performance is determined to be 0.483. Both values surpass the threshold of 0, indicating that the model exhibits good predictive relevance concerning these variables. Therefore, based on the obtained $Q^2$ values of 0.366 for personality traits and 0.483 for student academic performance, we can conclude that the predictive relevance of the model in this analysis is deemed suitable. The $Q^2$ value for student academic performance is the highest, suggesting that this variable has the strongest predictive power. These values suggest that the model can effectively predict and explain variations in personality traits and student academic performance.

**Table 4. Hypotheses testing of direct relationships.**

| | | Beta | Mean | STDEV | T value | P Values | 2.5% | 97.5% | Decision |
|---|---|---|---|---|---|---|---|---|---|
| H1 | Entrepreneurial intention>Personality Traits | 0.885 | 0.897 | 0.022 | 39.858 | 0.000 | 0.854 | 0.942 | Supported |
| H2 | Entrepreneurial Intention>Student Academic Performance | 0.083 | 0.107 | 0.168 | 0.495 | 0.621 | -0.228 | 0.431 | Not supported |
| H3 | Personality Traits>Student Academic Performance | 0.835 | 0.815 | 0.154 | 5.421 | 0.000 | 0.496 | 1.106 | Supported |
| H4 | Entrepreneurial intention>Personality Traits> Student Academic Performance | 0.738 | 0.731 | 0.141 | 5.231 | 0.000 | 0.440 | 1.022 | Supported |

**Table 5. Q² predictive relevance.**

|  | SSO | SSE | Q² (= 1-SSE/SSO) |
|---|---|---|---|
| EI | 750.000 | 750.000 | |
| PT | 750.000 | 475.380 | 0.366 |
| SAP | 750.000 | 387.542 | 0.483 |

**Table 6. F² effect size.**

| Constructs | F Square | p-value |
|---|---|---|
| EI | 3.603 | 0.051 |
| PT | 0.152 | 0.696 |
| SAP | 0.872 | 0.352 |

*5.4.2.3 F2 effect size.* The F2 value measures the explanatory power of a variable in a PLS-SEM model. It is calculated by dividing the explained variance by the residual variance [142]. The p-value is the probability of obtaining the observed F2 value by chance. In Table 6, the F2 value for EI is the highest, suggesting that this variable has the strongest explanatory power. The F2 values for PT and SAP are relatively small, suggesting that these variables do not explain a large amount of variance in the dependent variable. The p-values for all three variables are above 0.05, suggesting that the observed F2 values are not statistically significant. This means we cannot be confident that the observed relationships between the variables are not due to chance. The results of this F2 table suggest that the model does not explain a large amount of variance in the dependent variable. However, there are some significant effects between the variables. Further research may be needed to determine the causal relationships between the variables.

*5.4.2.4 Goodness of fit index.* The goodness of fit index (GFI) measures how well the model fits the data. A higher GFI indicates a better fit. The threshold value for a small effect is 0.10, a medium effect is 0.25, and a large effect is 0.36 [143]. In this example, the GFIs for all three constructs are above the threshold values for small effects. This suggests that the model fits the data well. However, the GFI for SAP is the highest, suggesting that this construct best fits the data. The effect size measures the strength of the relationship between a construct and the dependent variable. A higher effect size indicates a stronger relationship. Here, in Table 7, the effect sizes for all three constructs are above the threshold value for a small effect. This suggests that all three constructs have a significant relationship with the dependent variable. The results of this goodness of fit table suggest that the model fits the data well and that all three constructs have a significant relationship with the dependent variable.

**5.4.3 Discriminant validity (Attribute-based perceptual mapping).** Wilk's lambda is a statistical measure that ranges between 0 and 1, providing insights into the model's ability to discriminate between different groups. When Wilk's lambda value is closer to 0, it indicates a higher discrimination power of the model. In other words, a smaller lambda value suggests

**Table 7. Goodness of fit index.**

| Constructs | Goodness of Fit index | Threshold value |
|---|---|---|
| EI | 0.561 | 0.10- Small effect |
| PT | 0.625 | 0.25- Medium effect |
| SAP | 0.642 | 0.36- Large effect |

**Table 8. Wilks' lambda.**

| Test of Function(s) | Wilks' Lambda | Chi-square | df | Sig. |
|---|---|---|---|---|
| 1 through 2 | .651 | 39.699 | 36 | .309 |
| 2 | .850 | 15.029 | 17 | .593 |

**Table 9. Functions at group centroids.**

| Functions at Group Centroids | | |
|---|---|---|
| Pocket money | Function | |
| | 1 | 2 |
| 1 | 0.060 | 0.298 |
| 2 | 0.535 | -0.637 |
| 3 | -1.407 | -0.413 |

that the model can effectively distinguish and classify the groups under consideration. On the other hand, if the value is closer to 1, it implies a reduced discrimination power, indicating that the model struggles to differentiate between the groups. In Table 8, the Wilk's lambda value is calculated to be 0.850 in the specific case of the analysis conducted. This value falls closer to 1, which signifies the model's lower discrimination power. It suggests that the model might face challenges in accurately distinguishing and classifying the groups based on the examined variables or factors.

This has been drawn separately in Excel using the data from Table 9 (function group centroids for different pocket money) and Table 10 (standardized discriminant function coefficients of each attribute on each function), also plotted in Fig 4.

**Table 10. Standardized canonical discriminant function coefficients.**

| Standardized Canonical Discriminant Function Coefficients | | |
|---|---|---|
| | Function | |
| | 1 | 2 |
| ATE1 | 0.168 | -0.277 |
| ATE2 | 0.741 | -0.387 |
| ATE3 | -0.307 | 0.017 |
| PBC1 | -0.707 | -0.258 |
| PBC2 | -0.245 | -0.104 |
| PBC5 | 0.090 | 0.557 |
| SN1 | 0.466 | 0.230 |
| SN2 | 0.292 | 0.252 |
| SN3 | -0.074 | 0.336 |
| PES1 | -0.333 | -0.110 |
| PES5 | -0.301 | -0.132 |
| PES6 | 0.277 | -0.232 |
| PTA1 | 0.390 | -0.186 |
| EP2 | -0.686 | 0.217 |
| EP4 | 0.048 | 0.581 |
| EP5 | 0.416 | -0.866 |
| PTA3 | -0.081 | 0.088 |
| PTA4 | -0.079 | 0.322 |

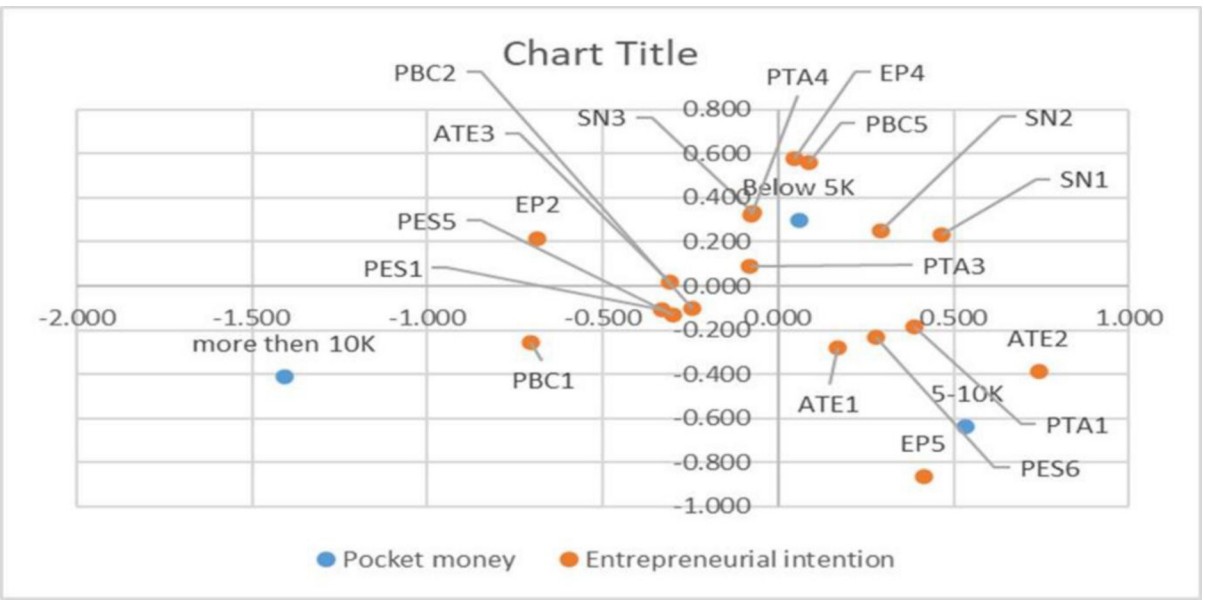

**Fig 4. Attribute-based perceptual map.**

The above attribute-based perceptual map in Fig 4 explains represent values of entrepreneurial intention are plotted using two distinct dimensions, with each discriminant function symbolizing a separate dimension. The first dimension consists of students receiving pocket money ranging from Rs. 5000 to Rs. 10000. This group prefers pursuing an entrepreneurial career over other options. They find the idea of being an entrepreneur appealing, and if they come across an excellent business opportunity, they exhibit a strong inclination to act upon it. Furthermore, their university actively supports students with entrepreneurial intentions by providing networking opportunities. Moreover, these students are inclined to seek employment where they can engage in creative and innovative activities.

Moving on to the second dimension, we have students who receive pocket money of less than Rs. 5000. Within this group, their primary motivation for pursuing a career as self-employed individuals lies in the opinions of their closest friends. They genuinely care about what their friends think when deciding to embark on a self-employed career. Like the first dimension, they find the prospect of being an entrepreneur attractive. When presented with a good business opportunity, they display eagerness and enthusiasm to seize it. Additionally, they prefer jobs that allow them to work independently.

Finally, the third dimension encompasses students who receive pocket money exceeding Rs. 10,000. These individuals are highly motivated and prepared to establish a viable business. They consider self-employment as a relatively effortless endeavor. They actively pursue a career as self-employed individuals and genuinely desire to start a business if provided with the necessary resources and opportunities. Their university recognizes their entrepreneurial aspirations, offers elective courses on entrepreneurship, and arranges conferences and workshops related to entrepreneurship.

## 6 Result and discussion

The findings of this study provide valuable insights into the relationship between entrepreneurial intention, personality traits, and student academic performance. The results indicate that personality traits mediate the relationship between entrepreneurial intention and student

academic performance, thus supporting H1. This suggests that the influence of entrepreneurial intention on academic performance is channelled through the individual's personality traits. Individuals with certain personality traits like employment preference and locus of control tend to exhibit better academic performance due to their entrepreneurial intentions.

However, the study also reveals that entrepreneurial intention alone does not significantly affect student academic performance, indicating that H2 is unsupported. This finding implies that while entrepreneurial intention may indirectly influence academic performance through personality traits, it does not directly impact academic performance. Other factors, such as the actual implementation of entrepreneurial activities or specific strategies, may be necessary to translate entrepreneurial intention into academic performance outcomes.

Furthermore, the study highlights the significant impact of personality traits on Student academic performance, providing support for H3. This suggests that certain personality traits positively influence academic performance among university students. Personality traits shape students' behaviors, study habits, and motivation, ultimately contributing to academic success.

Interestingly, the study also reveals that entrepreneurial intention substantially impacts student academic performance by mediating personality traits, supporting H4. This implies that individuals with higher entrepreneurial intentions tend to exhibit certain personality traits that positively influence their academic performance. The entrepreneurial intention may foster self-motivation, perseverance, and goal orientation, which, in turn, enhance academic performance outcomes.

The study's findings suggest that focusing more intensively on specific dimensions, particularly perceived behavioral control, educational support, and subjective norm, is critical to cultivating a robust entrepreneurial intention among students. Perceived behavioral control denotes the individual's confidence in their capability to execute entrepreneurial actions successfully, and the study underscores its pivotal role in shaping the intention to embark on entrepreneurial ventures. Similarly, the influence of perceived educational support, stemming from guidance and resources provided by educational institutions, emerges as a critical factor in nurturing students' inclination toward entrepreneurship. Furthermore, the subjective norm, which encapsulates the influence of social and peer expectations, plays a vital role in molding students' aspirations toward entrepreneurship.

Moreover, the study illuminates the potential of specific personality traits—notably, an emphasis on employment preference and a strong locus of control—to be precursors to successful entrepreneurial intentions. Students inclined towards entrepreneurship, who prioritize the pursuit of self-employment over traditional employment, demonstrate a mindset conducive to venturing into innovative and entrepreneurial pursuits. Similarly, those with a resolute locus of control who believe in one's capacity to influence outcomes through actions exhibit qualities aligned with effective entrepreneurial intentions.

These identified personality traits encompass a range of attributes that extend beyond entrepreneurial intentions and have profound implications for students' holistic development. Notably, the enhancement of abilities, encompassing cognitive and practical skills and academic competence, emerges as a direct outcome of these traits. Additionally, the cultivation of self-efficacy—the belief in one's capability to accomplish tasks—and adeptness in setting and pursuing goals can be attributed to nurturing these personality traits. Equally significant is honing time management skills, a trait closely linked to entrepreneurial success.

Ultimately, the study highlights the broader implications of these personality traits for students' overall academic performance. By cultivating these attributes, educational institutions can empower students to navigate the academic landscape more effectively, leading to an enhanced learning experience and improved performance. Therefore, the study's findings not

only contribute to understanding the factors shaping entrepreneurial intentions but also hold the potential to drive positive transformations in students' academic journeys.

Finally, the study highlights the multifaceted nature of students' entrepreneurial intentions based on their pocket money amounts through discriminant validity analysis. While financial circumstances differ, the appeal of entrepreneurship remains consistent across dimensions. The study highlights the influence of peers, university support, and individual motivations in shaping students' entrepreneurial aspirations and behaviors. This subtle understanding enhances strategies for nurturing and facilitating entrepreneurial activities among students, irrespective of their financial support from family members.

## 7 Limitations and future scope of the study

This study is not without limitations. Using a descriptive research design in this study allows for a detailed investigation of the mediating effect of personality traits in the relationship between entrepreneurial intentions and academic performance among university students. A comprehensive understanding of the variables can be obtained by employing various data collection methods and focusing on the specific time and location of the study. While the descriptive design does not establish causality, it provides valuable insights into the phenomenon. It is a foundation for future research and practical implications in entrepreneurship and education. Further limitations of the study are noted as follows:

### 7.1 Cross-sectional design

The study employs a cross-sectional design, capturing data at a single point in time. This limits the ability to establish causality and observe changes in the relationship over time. A longitudinal design would provide a more robust understanding of how personality traits mediate the relationship between entrepreneurial intentions and academic performance.

### 7.2 Self-report measures

The study relies on self-report measures subject to response biases such as social desirability or recall bias. Participants may provide answers that they perceive as favorable or reflect their ideal self-image, potentially affecting the accuracy of the data collected.

### 7.3 Single institution or context

Although the institution considered to conduct this study has a student population from all over the nation, it may restrict the generalizability of the findings to other settings. Different academic environments, cultural contexts, or educational systems may influence the relationship between entrepreneurial intentions, personality traits, and academic performance.

### 7.4 Mediating variable complexity

Personality traits, as a mediating variable, are complex constructs influenced by multiple factors such as genetics, environment, and individual experiences. The study may not capture the full range of factors affecting personality traits and their mediating effect on the relationship between entrepreneurial intentions and academic performance.

### 7.5 Measurement of entrepreneurial intentions

The study's measurement of entrepreneurial intentions may be based on self-reported intentions rather than actual behaviors related to entrepreneurship. While intentions are often used

as a proxy for behavior, there may be a discrepancy between individuals' stated intentions and their actual engagement in entrepreneurial activities.

Recognizing these limitations is crucial for understanding the scope and implications of the study's findings. Future research could address these limitations by utilizing more extensive and diverse samples, employing longitudinal designs, integrating with the objective measures of behaviors related to entrepreneurship, and considering a broader range of contextual factors that may influence the relationship between personality traits, entrepreneurial intentions, and academic performance.

## 8 Practical implications

The present study explored the mediating role of personality traits in the relationship between entrepreneurial intentions and academic performance among university students. The literature review highlighted that personality traits are essential to entrepreneurial intentions and academic performance. This study's findings suggest a significant relationship between entrepreneurial intentions, personality traits, and academic performance. Moreover, the mediating effect of personality traits was found to play a crucial role in the relationship between entrepreneurial intentions and academic performance. These findings are consistent with previous studies, which have established the link between personality traits, entrepreneurial intentions, and student academic performance.

The results of this study have significant implications for educational institutions, policymakers, and entrepreneurs. Educational institutions should develop entrepreneurship education programs targeting students' personality traits to enhance their entrepreneurial intentions and academic performance. Policymakers should also support implementing such programs to foster the growth of the entrepreneurial ecosystem. Furthermore, entrepreneurs can use this information to understand the personality traits essential for success in entrepreneurship and use it to select and develop their team members.

## 9 Conclusion

This study contributes to the existing literature by providing insights into the mediating role of personality traits in the relationship between entrepreneurial intentions and academic performance. The findings highlight the importance of considering personality traits as essential predictors influencing entrepreneurial intentions and academic performance. By recognizing the mediating effect of personality traits, this study emphasizes the need for educational institutions, policymakers, and entrepreneurs to consider these factors when designing entrepreneurship education programs and formulating policies to support the growth of the entrepreneurial ecosystem.

The results of this study imply that personality traits play a crucial role in shaping students' entrepreneurial intentions and subsequent academic performance. This suggests that students with specific personality traits are likelier to exhibit higher entrepreneurial intentions, which can positively influence their academic performance outcomes. By understanding the mediating role of personality traits, educators and policymakers can tailor their strategies to cultivate and nurture these traits among students, enhancing both their entrepreneurial intentions and academic achievements.

Furthermore, the implications of this study extend beyond the academic realm. The findings emphasize the importance of fostering an entrepreneurial mindset and developing key personality traits contributing to entrepreneurial success. This has implications for the broader entrepreneurial ecosystem, as individuals with strong entrepreneurial intentions are more likely to contribute to innovation, economic growth, and job creation. Policymakers and stakeholders in entrepreneurship development can utilize these findings to design initiatives and

policies that promote cultivating specific personality traits, ultimately leading to a more vibrant and robust entrepreneurial ecosystem.

## Supporting information

**S1 Dataset.**
(PDF)

## Author Contributions

**Conceptualization:** Smita Panda, Vasumathi Arumugam.

**Formal analysis:** Vasumathi Arumugam.

**Methodology:** Smita Panda, Vasumathi Arumugam.

**Supervision:** Vasumathi Arumugam.

**Visualization:** Smita Panda, Vasumathi Arumugam.

**Writing – original draft:** Smita Panda.

**Writing – review & editing:** Vasumathi Arumugam.

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
