## [Decision Letter · Decision Letter 0]

24 Aug 2023

PONE-D-23-21632Exploring the Mediating Effect of Personality Traits in the Relationship between Entrepreneurial Intentions and Academic Performance among Students.PLOS ONE

Dear Dr. A,

Thank you for submitting your manuscript to PLOS ONE. After careful consideration, we feel that it has merit but does not fully meet PLOS ONE’s publication criteria as it currently stands. Therefore, we invite you to submit a revised version of the manuscript that addresses the points raised during the review process.

ACADEMIC EDITOR:The article is interesting and novel. I invite you to make changes as recommended by reviewers. Looking forward to your revision. 

We look forward to receiving your revised manuscript.

Kind regards,

Remya Lathabhavan

Academic Editor

PLOS ONE

Journal Requirements:

"NO. The funders had no role in study design, data collection and analysis, decision to publish, or preparation of the manuscript." 

5. Please amend your list of authors on the manuscript to ensure that each author is linked to an affiliation. Authors’ affiliations should reflect the institution where the work was done (if authors moved subsequently, you can also list the new affiliation stating “current affiliation:….” as necessary).

6. Please ensure that you refer to Figure 1 and 2 in your text as, if accepted, production will need this reference to link the reader to the figure.

Reviewers' comments:

Reviewer's Responses to Questions

**Comments to the Author**

1. Is the manuscript technically sound, and do the data support the conclusions?

Reviewer #1: Yes

Reviewer #2: Yes

2. Has the statistical analysis been performed appropriately and rigorously? 

Reviewer #1: Yes

Reviewer #2: Yes

3. Have the authors made all data underlying the findings in their manuscript fully available?

Reviewer #1: Yes

Reviewer #2: Yes

4. Is the manuscript presented in an intelligible fashion and written in standard English?

Reviewer #1: Yes

Reviewer #2: Yes

5. Review Comments to the Author

Reviewer #1: The paper's title is appropriate, and the abstract accurately summarizes its content. In the abstract and introduction sections, the author clearly states the goals and parameters of the work. The author's succinct summary of the literature reveals strong topic expertise. Additionally, it is cited with references from recent publications  as well. The processes for the research study are reasonable, appropriate, and well-explained. The results have been accompanied with encouraging descriptive statistics, and the connections between the research variables are afterwards analyzed. Future scholars may use the insights to create fresh methods for conducting research across diverse fields.

Reviewer #2: The subject matter is highly intriguing, and I found it to be quite captivating. I wish to thank you for your dedicated efforts in presenting your research endeavor with an exemplary level of professionalism. The language employed in the manuscript is commendable. The paper provides a thorough exploration of the existing literature, encompassing not just recent publications but also those from earlier periods. A substantial portion of relevant works on this subject has been aptly cited, seamlessly integrating them into the contextual framework. I have not identified any significant omissions of pertinent contributions. On the contrary, the employed literature showcases a commendable diversity.

The methodology utilized for the research is distinctly elucidated, underscoring its inherent value to the study. This is well-balanced by harmonizing theoretical and conceptual insights with the exploratory dimension of the research. The presentation and subsequent discourse of the results, along with the ensuing recommendations, greatly enhance the paper's practical relevance within the discipline. The applicability of these insights in real-world scenarios is evident. Furthermore, the contextual exploration of arguments vis-à-vis entrepreneurial intention among students lends an added layer of significance to the paper.

The conclusion adeptly encapsulates the pivotal arguments and findings. The conscientious recognition of limitations is noteworthy, and the proposed trajectory for future research enriches the paper's depth. The manuscript imparts thought-provoking insights, leading me to lean towards its publication.

However, prior to advancing a recommendation or acceptance, a few salient observations warrant attention to enhance both the work's quality and its alignment with the standards of this esteemed journal. I eagerly await the revised manuscript. I encourage the authors to thoroughly consider and incorporate the suggested articles to enhance the manuscript's quality, substantiating these choices as needed.

To further refine the manuscript, several aspects merit detailed consideration:

• The introduction section would benefit from a few additional sentences to fortify its foundation. Additionally, elucidating the rationale behind introducing the concept of "entrepreneurial behavior" in the introduction, considering its absence from subsequent discussion and potential irrelevance to the study's context, is advisable.

• Within the literature review, the incorporation of more recent works, particularly spanning the years 2020 to 2023, would undoubtedly heighten the manuscript's value.

• In the discussion, succinctly encapsulating the novelty and significance of the primary discovery into a concise and groundbreaking assertion would be advantageous.

• While the core arguments are lucidly expounded upon, certain paragraphs could benefit from refinement. Meticulous editing could significantly amplify the value of the content. Notably, instances of non-academic language and grammatical errors are discernible. Hence, a thorough editing pass is recommended, either by the author or a specialist, to ensure consistent use of academic language and the effective conveyance of ideas.

Commitment to these revisions will undoubtedly elevate the manuscript's overall quality, rendering it more apt for publication in this esteemed journal.

6. PLOS authors have the option to publish the peer review history of their article (what does this mean?). If published, this will include your full peer review and any attached files.

Reviewer #1: **Yes: **Dr. Niyati Ravi Patel

Reviewer #2: No

---

## [Author Response · Author response to Decision Letter 0]

29 Sep 2023

Dear Sir/Madam,

We have incorporated the comments given by the reviewers in our revised article. Kindly do the needful.

Thanks and Regards,

Authors

---

## [Decision Letter · Decision Letter 1]

10 Oct 2023

Exploring the Mediating Effect of Personality Traits in the Relationship between Entrepreneurial Intentions and Academic Performance among Students.

PONE-D-23-21632R1

Dear Dr. Arumugam,

We’re pleased to inform you that your manuscript has been judged scientifically suitable for publication and will be formally accepted for publication once it meets all outstanding technical requirements.

Kind regards,

Remya Lathabhavan

Academic Editor

PLOS ONE

Additional Editor Comments (optional):

Authors incorporated the suggested changes positively.

Reviewers' comments:

Reviewer's Responses to Questions

**Comments to the Author**

1. If the authors have adequately addressed your comments raised in a previous round of review and you feel that this manuscript is now acceptable for publication, you may indicate that here to bypass the “Comments to the Author” section, enter your conflict of interest statement in the “Confidential to Editor” section, and submit your "Accept" recommendation.

Reviewer #1: All comments have been addressed

Reviewer #2: All comments have been addressed

2. Is the manuscript technically sound, and do the data support the conclusions?

Reviewer #1: Yes

Reviewer #2: Yes

3. Has the statistical analysis been performed appropriately and rigorously? 

Reviewer #1: Yes

Reviewer #2: Yes

4. Have the authors made all data underlying the findings in their manuscript fully available?

Reviewer #1: Yes

Reviewer #2: Yes

5. Is the manuscript presented in an intelligible fashion and written in standard English?

Reviewer #1: Yes

Reviewer #2: Yes

6. Review Comments to the Author

Reviewer #1: (No Response)

Reviewer #2: The author has taken heed of the provided instructions and skillfully bolstered the introductory section to strengthen its underlying framework. Furthermore, in response to the inquiry regarding the introduction of entrepreneurial behavior, the author has meticulously substantiated this aspect within the manuscript itself, going the extra mile to reiterate their rationale in the accompanying cover letter. I must express my satisfaction with the author's thorough commitment to addressing this particular comment.

The inclusion of recent literature spanning the years 2020-2023 has been a valuable enhancement. I am appreciative of the authors for their diligence in substantiating this modification.

The authors have significantly expanded upon their discussion, providing an extensive elaboration of the novelty and significance of their primary discovery, which has now been distilled into concise yet groundbreaking assertions. I am thoroughly pleased with the way they have highlighted the unique aspects of the study.

Having thoroughly reviewed the manuscript, it is evident that the quality and coherence of the content have been notably elevated. Moreover, I've observed substantial improvements in the language employed throughout the text, contributing to a more polished and refined overall presentation.

7. PLOS authors have the option to publish the peer review history of their article (what does this mean?). If published, this will include your full peer review and any attached files.

Reviewer #1: No

Reviewer #2: No

---

## [Editor Report · Acceptance letter]

30 Oct 2023

PONE-D-23-21632R1 

Exploring the Mediating Effect of Personality Traits in the Relationship between Entrepreneurial Intentions and Academic Performance among Students. 

Dear Dr. Arumugam:

I'm pleased to inform you that your manuscript has been deemed suitable for publication in PLOS ONE. Congratulations! Your manuscript is now with our production department. 

Kind regards, 

on behalf of

Dr. Remya Lathabhavan 

Academic Editor

PLOS ONE